# Architecture of Nanoantioxidant Based on Mesoporous Organosilica Trp-Met-PMO with Dipeptide Skeleton

**DOI:** 10.3390/ma16020638

**Published:** 2023-01-09

**Authors:** Wanli Zhou, Haohua Ma, Yunqiao Dai, Yijing Du, Cheng Guo, Jianqiang Wang

**Affiliations:** School of Chemistry and Molecular Engineering, Nanjing Tech University, 30 Puzhu South Road, Nanjing 211816, China

**Keywords:** mesoporous organosilica, ROS, nanoantioxidant, tryptophan–methionine, dipeptide framework

## Abstract

A nanoantioxidant of mesoporous organosilica (Trp-Met-PMO) based on the framework of tryptophan–methionine dipeptide was first designed and constructed by condensation between self-created dipeptide organosilica precursor (Trp-Met-Si) and tetraethyl orthosilicate (TEOS) in alkaline conditions under the template hexadecyl trimethyl ammonium bromide (CTAB). Trp-Met-Si was prepared by the reaction between dipeptide Trp-Met and conventional organosilicon coupling agent isocyanatopropyltriethoxysilane (IPTES) via a multiple-step reaction method. The material Trp-Met-PMO was confirmed by XRD, FT-IR and N_2_ adsorption–desorption analysis. The material Trp-Met-5-PMO with low amounts of organosilica precursor remained a mesoporous material with well-ordered 2D hexagonal (*P6mm*) structure. With increasing amounts of organosilica precursor, a mesoporous structure was still formed, as shown in the material Trp-Met-100-PMO with the highest amounts of organosilica precursor. Moreover, pore size distribution, surface area and porosity of Trp-Met-PMO are regulated with different amounts of organosilica precursor Trp-Met-Si. The antioxidant activity of Trp-Met-PMO was evaluated by ABTS free radical-scavenging assay. The results showed that antioxidant activity was largely enhanced with increasing contents of organosilica precusor Trp-Met-Si in the skeleton. The material Trp-Met-40-PMO exhibited maximum scavenging capacity of ABTS free radicals, the inhibition percent was 5.88%. This study provides a design strategy for nanoantioxidant by immobilizing short peptides within the porous framework of mesoporous material.

## 1. Introduction

Reactive oxygen species (ROS) are usually produced in the process of cell metabolism due to incomplete reduction of oxygen molecules, and have different forms such as superoxide anion, hydrogen peroxide, hydroxyl radicals, etc. [1,2,3,4]. If high concentrations of ROS are generated in metabolism, they will result in oxidative stress and damage the surrounding normal tissues to produce diseases because these oxygen radicals affect proteins and lipids [5]. In order to moderate the overproduction of ROS and oxidative stress, some antioxidants are developed through directly scavenging ROS or indirectly mobilizing oxidative defense and inhibiting ROS production in many fields. Moreover, they also have the ability to form new and more stable free radicals through intramolecular hydrogen bonding and further oxidation. Thus, antioxidant materials or compounds are developed to decrease excess ROS, attracting the attention of many researchers.

With the remarkable breakthroughs in the field of nanochemistry and nanofabrication technology, various nanomaterials have been prepared to regulate amounts of ROS for extensive biomedical applications. Specially, ROS-scavenging nanomaterials display significant advantages based on the different formations including antioxidant drug-loaded nanoparticles and ROS-responsive nanomaterial without additions [6,7,8]. When designing these nanomaterials, many factors should be considered because they are tightly related to the application field and environment. For example, the morphology, surface areas, geometrical features and intrinsic characteristics directly influence the ROS-regulating performance [9]. Among these materials, mesoporous silica nanomaterials (MSNs) have received more attention because of tunable porosity, large specific surface area, low density, thermal insulation, high stability of solvents and easy surface functionalization [10,11]. Moreover, they can usually be used as nanocarriers of drugs due to their good biocompatibility. Through linking ROS-responsive moieties, MSNs are designed to develop ROS-responsive and ROS-scavenging materials in different fields. For example, metformin (MET) was loaded on MSN, which provided the possibility for long-term ROS-scavenging activity to achieve the antioxidative effect [12]. Methylthio-functional MSN was prepared to regulate amounts of ROS by the oxidation of methylthiopropyl groups to sulfoxides in response to the presence of H_2_O_2_ [13]. Two antioxidant enzymes were simultaneously delivered by functional MSNs, which revealed synergistic efficiency of ROS scavenging [14]. Thus, MSNs have become a popular choice for modulating excessive ROS.

Many studies have confirmed that free amino acids and peptides derived from plants, animals, traditional foods and biological metabolisms exhibit good antioxidant activities [15,16,17]. Actually, the antioxidant activities of peptides are mainly attributable to their antioxidant amino acid residues, which are reported in many studies [18,19]. Among 20 protein-derived amino acids, tryptophan and methionine indicate high reactive activities in scavenging ABTS+, which reveals that they possess antioxidant activity because of the indole NH group of Trp and thioether group of Met [20]. When combining dipeptide and MSNs without destroying the antioxidant activity of dipeptide, some previous studies indicated that the main strategy was a post-grafting method by adsorption of dipeptide or amino acid on the mesoporous materials [21,22]. In order to improve immobilization stability of these active molecules and maintain the antioxidant activity of dipeptide, they could be used as skeleton components to form porous channel in the co-condensation process.

Based on the above discussion, in the current study, MSN with ROS-scavenging ability was prepared by immobilizing ROS-scavenging moieties within the porous skeleton without additional molecules by a co-condensation method. Furthermore, the biomolecule dipeptide tryptophan–methionine (Trp-Met) was first selected as a ROS-scavenging moiety to construct MSNs. The organosilicon precursor of dipeptide Trp-Met was synthesized by the reaction between Trp-Met and conventional alkoxysilane precursor 3-isocyanate propyltriethoxysilane (IPTES). Through regulating the ratios of dipeptide organosilicon precursor during preparation, mesoporous organosilica materials with various contents of dipeptide were obtained. They were characterized by XRD, FTIR and nitrogen adsorption–desorption analysis. We monitored ROS-scavenging ability of these materials by an ABTS free radical-scavenging experiment. The mesoporous organosilica was confirmed as a potential antioxidant to scavenge excessive ROS because of dipeptide Trp-Met.

## 2. Materials and Methods

### 2.1. Materials

Methionine, tryptophan and tetraethyl orthosilicate (TEOS) were purchased from Aladdin. 2,2′-Azino-bis-(3-ethylbenzothioazoline-6-sulfonic acid) (ABTS), (3-Isocyanatopropyl) triethoxysilane (IPTES), N-Boc-ethylenediamine, O-(Benzotriazol-1-yl)-N,N,N′,N′-tetramethyluroniumtetrafluoroborate (TBTU) and di-tert-butyl dicarbonate (BOC_2_O) were purchased from Energy. CTAB was purchased from Adamas-beta. Other common reagents and solvents were AR grade without further purification.

### 2.2. Characterization

^1^H-NMR spectra were recorded in DMSO-*d*_6_ on a Bruker AC-400 Avance (400 MHz) spectrometer at 290 K. All chemical shifts (ppm) were reported in the standard notation of parts per million (δ) relative to tetramethylsilane (TMS). The mass spectrum was measured on an Agilent 6550 iFunnel Q-TOF with chemical ionization. UV–Vis adsorption spectra were recorded on a Shimadzu 1800 UV/Vis spectrophotometer to evaluate the ABTS radical-scavenging experiment. FTIR spectra were performed on a Thermo Nicolet iS5 FT instrument in the mid-infrared region of 4000–400 cm^−1^ using a KBr beam-splitter. The samples were mixed with a ratio of approximately 1:100 (samples/KBr, *w*/*w*) and pressed into a thin and transparent disk. N_2_ adsorption–desorption isotherms were measured on a Micrometrics ASAP2020. Before the measurements, all the materials were degassed at 150 °C for 6 h. The specific surface area (S_BET_) was calculated based on the Brunauer–Emmet–Teller (BET) method. The Barret–Joyner–Halenda (BJH) method was used for calculations of pore size distributions. Wide-angle X-ray diffraction (XRD) patterns were acquired using a D/Max-2400 Powder X-ray diffractometer with a Cu Kα X-ray radiation source (λ = 1.5418 Å).

### 2.3. Synthesis of Dipeptide Organosilica Precursor (Trp-Met-Si)

#### 2.3.1. Synthesis of Dipeptide BOC-Trp-Met-OMe

The compound of tryptophan–methionine dipeptide organosilica precursor (Trp-Met-Si) was prepared via a multiple-step reaction, as shown in Figure 1I. First, the amino group of Trp and carboxyl group of Met were protected by different compounds to our previous study on the preparation of dipeptide [23]: A 250 mL round-bottom flask was charged with tryptophan (5.97 g, 20 mmol) in 50 mL of sodium hydroxide solution (1 M). Then, the solution of di-tert-butyl dicarbonate (4.8 g, 22 mmol) in 50 mL tetrahydrofuran was slowly added into the above solution containing tryptophan at 0 °C. The reactive mixture was stirred at room temperature overnight. Subsequently, 1 M HCl was used to change the pH of the reactive solution to 2. After completion of the reaction, the reaction mixture was extracted with ethyl acetate. The organic phase was washed with saturated salt water and dried with anhydrous sodium sulfate. Then, it was concentrated under reduced pressure to obtain the product. Finally, 4.26 g (14 mmol, 70 %) of N-(tert-butoxycarbonyl)-tryptophan (designated as BOC-Trp) was obtained as a white solid.

Thionyl chloride (9 mL, 120 mmol) was slowly added into a clear solution of methionine (5.97 g, 40 mmol) in 100 mL methanol at 0 °C. The reaction mixture was placed in a 250 mL round-bottom flask at 80 °C for 8 h. After that, the methanol was removed under reduced pressure by vacuum rotary evaporation. The dichloromethane was added into the resulting mixture, and the excess SOCl_2_ was removed to obtain the product. Finally, 5.75 g (35 mmol) of methionine methyl ester hydrochloride (designated as Met-OMe) was obtained with a yield of 87% as a white solid. 

Subsequently, the compounds of TBTU (7.68 g, 24 mmol) and triethylamine (8.5 mL, 60 mmol) were added into the 100 mL anhydrous THF solution of BOC-Trp (6.08 g, 20 mmol) at 0 °C. The reaction mixtures were kept at the same temperature and stirred for 30 min. Whereafter, the compound Met-OMe (3.99 g, 20 mmol) was added into the reaction system and stirred at 0 °C for 10 min. After the completion of the reaction, the reaction systems were raised to 25 °C and continuously stirred for 3 h. The solvent THF was removed by vacuum rotary evaporation under reduced pressure. The crude product was extracted with ethyl acetate, and the organic phase was washed with saturated sodium bicarbonate and saturated salt water and dried with anhydrous sodium sulfate. Finally, 6.74 g (15 mmol) of BOC-Trp-Met-OMe was obtained with a yield of 75% as a white solid after removing the solvent.

#### 2.3.2. Synthesis of Dipeptide Organosilica Precursor Trp-Met-Si

BOC-Trp-Met-OMe (6.74 g, 14 mmol) was added to 50 mL of 1 M sodium hydroxide solution and tetrahydrofuran in a 250 mL round-bottom flask, stirred at 25 °C for 2 h and then tetrahydrofuran was removed by vacuum rotary evaporation. The pH of the solution was adjusted to 2 with 1 M hydrochloric acid, and the crude product was extracted with ethyl acetate. The obtained organic phase was washed with saturated brine and dried with anhydrous sodium sulfate. Then, the solvent was removed by vacuum rotary evaporation. Finally, 4.36 g (10 mmol) of white solid product (designated as BOC-Trp-Met) was obtained with a yield of 70%.

The compound BOC-Trp-Met (3.2 g, 7.4 mmol) was dissolved in 100 mL of dried tetrahydrofuran. EDCl (1.69 g, 8.8 mmol) and NHS (1.02 g, 8.8 mmol) were added into the above reaction system at 0 °C. The resulting mixtures were stirred at 0 °C for 1 h and then raised to 25 °C for 5 h. After that, N-Boc-Ethylenediamine (1.18 g, 7.4 mmol) and an appropriate amount of triethylamine were added and stirred at 25 °C for 5 h. The obtained product was purified by silica gel column chromatography. Finally, 4.04 g (7 mmol) of white solid powder was obtained with a yield of 87%. The obtained product was dissolved in 50 mL of methanol. Five milliliters of acetyl chloride was added into the solutions of methanol at 0 °C, and the resulting mixtures were heated to 25 °C and stirred for 2 h. The solvent was removed by vacuum rotary evaporation, and the colorless oily substance was obtained in a bottle. Then, the oily substance was dissolved in 100 mL of dried tetrahydrofuran. Triethylamine (1.34 g, 13.2 mmol) and IPTS (3.27 g, 13.2 mmol) were added and stirred for 5 h at room temperature. After the solvent was removed under reduced pressure, the product was washed with n-hexane and naturally dried at room temperature. Finally, 4.33 g (5.5 mmol) of white solid product dipeptide organosilica precursor (designated as Trp-Met-Si) was obtained with a yield of 71%.

### 2.4. Preparation of Mesoporous Organosilica (Trp-Met-PMO)

The synthesis of ordered mesoporous organosilica with the framework of methionine–tryptophan dipeptide is shown in Figure 1II. The mesoporous organosilica Trp-Met-Si was prepared by co-condensation of TEOS and Met-Trp-Si under the template CTAB with the following process: The surfactant CTAB was added into a mixture of 1 M sodium hydroxide solution and H_2_O in a 250 mL beaker. The resulting mixture was stirred at room temperature for 30 min to form a transparent solution. Then, the beaker was transferred to a water bath of 40 °C. Different proportions of Trp-Met-Si and TEOS were slowly added into the beaker and vigorously stirred for 24 h. The obtained product was washed with distilled water and dried at room temperature. Finally, the template CTAB was removed by extraction with mixed solutions of 5 mL concentrated hydrochloric acid and 250 mL EtOH at 85 °C for 48 h. The obtained material is denoted as Trp-Met-x-PMO (x = 5, 10, 15, 20, 40, 60, 100, where x is the molar percentage of Trp-Met-Si/(Met-Trp-Si + TEOS)).

### 2.5. ABTS Radical-Scavenging Experiment

The scavenging activity of mesoporous organosilica Trp-Met-PMO on hydrophilic ABTS free radicals was evaluated according to the method in the literature. ABTS was dissolved in 7 mm water. ABTS radical cation (ABTS+) was produced by the reaction of ABTS stock solution with 2.45 mM potassium persulfate, which was used after the mixed solutions were placed in the dark at room temperature for 12–16 h. The reaction ratio of ABTS to potassium persulfate was 1:0.5, which resulted in incomplete oxidation of ABTS. The oxidation reaction of ABTS began immediately, but the absorbance did not reach the maximum value until 6 h later and tended to be stable. Finally, the concentrations of ABTS+ were adjusted to give absorbance of 0.970 ± 0.020 at 734 nm. Free radicals in this form were still stable when they were stored in darkness at room temperature for more than two days.

To evaluate the scavenging effect of the material Trp-Met-PMO on ABTS, 20 mg of Trp-Met-PMO was mixed with 25 mL of ABTS radical solution. The obtained mixtures were kept away from light for 5 min at 37 °C. The decrease in absorbance at 734 nm was measured at the end of 5 min. Antioxidant activity was expressed as a percentage of ABTS free radical scavenging, as shown in the following equation:Inhibition (%) = (Ao − A1)/Ao
where Ao is the absorbance of the initial concentration of ABTS (mg/L) without the sample. A1 is the absorbance of the concentration of ABTS at the end of 5 min (mg/L) in the presence of the sample.

## 3. Results and Discussion

The mesoporous organosilica Trp-Met-PMO was prepared by hydrolysis and co-condensation of two different silicon sources under the template, indicated in Figure 1II. One organic silicon source of organosilica precursor Trp-Met-Si was first designed and synthesized by using Trp-Met dipeptide as the main reactant via a multiple-step method, as shown in Figure 1I. The molecule Trp-Met-Si was confirmed by ^1^H NMR, ^13^C NMR and ESI-MS (seen in Appendix A). Meanwhile, the ratios of Trp-Met-Si and TEOS were regulated to obtain the material Trp-Met-PMO with Trp-Met-Si within the skeleton. Moreover, the material Trp-Met-100-PMO was produced by using the only organic silicon source Trp-Met-Si to accomplish functional component immobilized on the whole porous framework. 

The Fourier transform infrared spectrum (FTIR) is used to illustrate the chemical bonds of mesoporous organosilica Trp-Met-PMO in Figure 1. The material Trp-Met-PMO containing different amounts of dipeptide Trp-Met of 5, 10, 15, 20, 40, 60, 100 exhibited similar FTIR. As shown in Figure 1, the N–H stretching vibration peaks of all the materials were observed at the wavenumber of 3303 cm^−1^. The wavenumber of 2980 cm^−1^ is the stretching vibration peak of the carbon–hydrogen bond (SC-H) from the thioether group. The wavenumbers of 1700 cm^−1^ and 1680 cm^−1^ denoted the stretching vibration peaks of the carboxyl group (C=O) from ester and amide, respectively. The stretching vibration peak of the sulfur–carbon bond (S-CH) from the thioether group was observed at the wavenumber of 1259 cm^−1^. These characteristic peaks were related to dipeptide organosilica precursor Trp-Met-Si. These results indicated that the compound Trp-Met-Si was successfully immobilized within the material Trp-Met-PMO. Moreover, the intensity of these peaks was enhanced with the increasing amounts of Trp-Met-Si. In addition, the stretching vibration peaks of Si-O-Si were observed at the wavenumbers of 1082 cm^−1^, 798 cm^−1^ and 462 cm^−1^. The noncondensing Si-OH peaks appeared at the wavenumber of 960 cm^−1^. These characteristic peaks illustrated that the material Trp-Met-PMO remained a mesoporous skeleton based on the condensation of orgnosilica precursor Trp-Met-Si and TEOS under the template.

The wide-angle X-ray diffraction (XRD) patterns of mesoporous organosilica Trp-Met-x-PMO are shown in Figure 2. All the materials exhibited a high intensity peak from the (100) plane in the range of 2θ from 1° to 7°. Moreover, the intensity of the diffraction peak varied with the changing amounts of dipeptide Trp-Met in the skeleton of the mesoporous organosilica material. Among these materials, only the sample Trp-Met-5-PMO with low amounts of Trp-Met showed three diffraction peaks corresponding to a strong intensity peak of (100) and two weak intensity peaks of (110), (200), which indicated a relatively broad diffraction peak. The result indicated that the material Trp-Met-5-PMO remained a mesoporous material with well-ordered 2D hexagonal (*P6mm*) structure similar to MCM-41. When the amounts of Trp-Met were increased in the preparation of mesoporous organosilica Trp-Met-PMO, two diffraction peaks of (110) and (200) became unapparent and even disappeared in XRD patterns. However, the diffraction peak of (100) was still observed for all of the materials. The interplanar spacing (*d*_100_) and cell parameters (α_0_) also changed with the variation of 2θ values, as shown in Table 1. In addition, the material Trp-Met-100-PMO also displayed the diffraction peak (100) with relatively low intensity without the addition of inorganic silicon source TEOS, which illustrated that the mesoporous structure was still formed by using organic silica precursor Trp-Met-Si as the only silicon source under the template. Furthermore, as shown in Figure 2, the intensity of diffraction peaks became weak with the increasing amounts of organosilica precursor Trp-Met-Si, which was due to the reduction of scattering ability from the disturbance of organosilica precursor within the framework.

Nitrogen adsorption–desorption analysis was performed to observe the pore structure of the materials Trp-Met-x-PMO (x = 5, 10, 15, 20, 40, 60, 100). Figure 3 shows N_2_ adsorption–desorption isotherms of the material Trp-Met-PMO with the different contents of Trp-Met-Si. All the materials exhibited type IV isotherms with sharp capillary condensation steps at a relative pressure P/P_0_ range between 0.8 and 1.0. Different types of hysteresis loops were observed in the materials with variable amounts of organosilica precursor Trp-Met-Si. The materials Trp-Met-5-PMO, Trp-Met-10-PMO, Trp-Met-15-PMO and Trp-Met-20-PMO indicated *H1*-type hysteresis loops, the other three materials presented *H3*-type hysteresis loops with the characteristic of narrow pore size distributions. These results illustrated that the material Trp-Met-PMO maintained a structure similar to ordered mesoporous silica material after immobilizing organosilica precursor Trp-Met-Si within the skeleton.

In addition, the structural parameters of the materials including BET specific surface area (S_BET_), BJH pore diameter (Dp), total pore volume (Vt) and wall thickness (W_T_) are listed in Table 1. With the increasing amounts of organosilica precursor, these parameters, except for the wall thickness, decreased, which confirmed that the organosilica precursor Trp-Met-Si influenced the surface properties of the materials. Moreover, these results further indicated that organosilica precursor Trp-Met-Si was immobilized on the skeleton of the materials. Meanwhile, functional groups of the molecule Trp-Met-Si were partially extended within the channels of the mesoporous organosilica materials.

The ABTS-scavenging assay was determined according to the literature with a slight modification in order to evaluate antioxidant properties of mesoporous organosilica material Trp-Met-PMO. Figure 4 shows the ABTS radical-scavenging rate of these materials. It can be seen that the ability of ABTS free radical scavenging was enhanced with the increasing content of organosilica precursor Trp-Met-Si in the range of 5 to 60. When the content of organosilica precursor Trp-Met-Si reached 40%, the inhibition of these materials slowed down, which indicated that their antioxidant capacity became weak. When the material Trp-Met-100-PMO was prepared by using Trp-Met-Si as the only silicon source, its antioxidant capacity decreased. These results illustrated that organosilica material Trp-Met-PMO possessed antioxidant property without the addition of the other compound because tryptophan–methionine dipeptide provided a certain amount of reducibility. Moreover, its antioxidant ability was regulated by changing amounts of Trp-Met-Si within the skeleton. In addition, the porous structure and structural parameters of the material were tightly related to antioxidant property, which resulted in an impact on oxidation resistance.

In order to observe the relation between the concentrations of the material Trp-Met-PMO and the ABTS free radical scavenging, Trp-Met-20-PMO with different concentrations was selected to perform the experiment. The results are indicated in Appendix A. Low concentrations of Trp-Met-PMO in the solution showed weak antioxidant ability. When the concentration was 0.8 mg/mL, the scavenging rate of ABTS free radicals reached the maximum and tended to be unchanged with increasing concentrations of the material. It is foreseen that the material Trp-Met-PMO can be effectively used as antioxidant material without adsorption of the substance.

## 4. Conclusions

In summary, the mesoporous organosilica material Trp-Met-PMO could be readily prepared by the co-condensation of organosilica precursor Trp-Met-PMO and TEOS under the template CTAB. Moreover, amounts of Trp-Met-Si were regulated to observe the changing of their structural features and antioxidant properties. For example, pore size distribution (from 2.4 nm to 1.2 nm), surface area (from 59 m^2^/g to 19 m^2^/g) and porosity (from 0.54 cm^3^/g to 0.06 cm^3^/g) of Trp-Met-PMO can be controlled by varying the molar percentage of Trp-Met-Si from 5 to 100. The material Trp-Met-5-PMO with low amounts of organosilica precursor remained a mesoporous material with well-ordered 2D hexagonal (*P6mm*) structure. The mesoporous skeleton could still be maintained in these materials even if Trp-Met-PMO was used as the only silicon source. In addition, these materials with different contents of Trp-Met-Si exhibited good antioxidant properties in the ABTS free radical-scavenging experiment. The material Trp-Met-40-PMO exhibited maximum scavenging capacity of ABTS free radicals, the inhibition percent was 5.88%. Based on these results, the material Trp-Met-PMO could probably be applied in cosmetics as an antioxidant additive in future. Meanwhile, the mesoporous organosilica Trp-Met-PMO may provide some advantages in biological systems with the combination of active molecules due to the dipeptide in the framework being the main component. 

## Data Availability

Not applicable.

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
