# Peer review of "Architecture of Nanoantioxidant Based on Mesoporous Organosilica Trp-Met-PMO with Dipeptide Skeleton"

_materials, 2023, doi:10.3390/ma16020638_

Round 1
Reviewer 1 Report
The article Architecture of novel nano-antioxidant based on mesoporous 2 organosilica Trp-Met-PMO with dipeptide skeleton Is accepted after consideration of the following major comments.
1) the full name of abbreviations should be mentioned where firstly appeared for example in abstract line 10 TEOS
2) Abstract, the results should be displayed and compare with positive standard.
3) Abstract; conclusion should add
4) line 66 should merge with 67
5) the literature survey about combination between dipeptide and mesoporous organosilica should mention and rational for this study should improve.
6) Synthesis of dipeptide is complicated method and should divide to steps.
7) The authors should mention positive standard.
8) Conclusion should improve and indicting the potential of this study.
9) References should be updated
Author Response
Dear reviewer:
Thank you very much for your attention and evaluation on our paper " Architecture of nano-antioxidant based on mesoporous organosilica Trp-Met-PMO with dipeptide skeleton (Ms. No.:materials-2064778)”. We have revised the manuscript according to your kind advice and detailed suggestions. The revised parts are marked with blue font. Enclosed please find the response to you.
Sincerely yours!
Jianqiang Wang

Reviewer 2 Report
This manuscript is dedicated to the preparation of an architecture of nano-antioxidant based on mesoporous organosilica Trp-Met-PMO with dipeptide skeleton. The authors employ an original preparation process, while they claim notable antioxidant activity of Trp-Met-PMO which is enhanced with the increasing contents of organosilica precusor Trp-Met-Si in the mentioned skeleton. The authors characterize the material by employing XRD, FT-IR and N2 adsorption-desorption analysis whereby all characterization results are particularly indicative. And the characterization effort, all in all, is much adequate and well-planned for the purposes of the present task.
Also, the authors discuss, systematize, and clarify the important aspects for such a material system which is not well-studied yet. More broadly the authors provided valuable ideas for the design of nano-antioxidant based on mesoporous material with framework of biomolecule dipeptides. The discussion provided is adequate for the present purpose. The well-detailed and at the same time comparative context of the experimental/characterization results clarifies convincingly the authors’ conclusions.
From practical point of view, the reported results thus bring new knowledge and certainly represent an original contribution in the present context.
The authors chose an adequate structure of the manuscript – an excellent point of departure for such a study. Also, they provided a balanced realistic and nicely illustrated presentation of their results and corresponding analysis that is of much scientific and practical interest and adds new knowledge to the field.
The present manuscript is a significant contribution, this work once published would be instructive and suggestive in terms of further studies and to a wider readership.
There are some relatively minor issues with this already excellent manuscript that will need to be addressed before the manuscript becoming suitable for publication, i.e., it can be considered for publication after a minor revision:
1: Title should not contain adjectives like “novel”, “excellent”, “original”, etc. These aspects are revealed by the results and not by statements. The same apply to the abstract, and to a large extent to main text too. Please, remove such declarative adjectives.
2: Authors are a little bit too telegraphic in what concerns the important fact that particle size distribution, surface area and porosity of Trp-Met-PMO are controllable: This should be mentioned already in the abstract. It will publicize their work better if they are more detailed/literal concerning this issue (in the abstract), while in the conclusion this controllability should be supported by quantitative ranges.
3: Is there any data about the thermal stability of the white solid dipeptide organosilica precursor (designated as Trp-Met-Si), this aspect should be also addressed in the text more clearly.
4: In the introduction, the authors miss that precursors with similar complexity and with similar interest concerning the interplay between design technique and properties to those achieved in the present work have been addressed and guided by reliable high-level ab initio theoretical methods of simulation. Examples in which such theoretical works help understanding synergies between morphology and design, and help addressing experimental challenges include: Dalton Transactions 44 (2015) 3356-3366, and Spectrochimica Acta Part A: Molecular and Biomolecular Spectroscopy 245 (2021) 118939. Such works should be referenced to.
5: In the context of the present discussion of local chemical bonding, e.g., of the sulfur-carbon bonds, stating bond lengths may be of interest to the readers? Such data may corroborate the present results.
6: Spell-check and stylistic revision of the paper are still necessary. Some long sentences, misspellings, etc., still are noticeable throughout the text.
Author Response

(The authors gave the same response as above.)

Round 2
Reviewer 1 Report
the article is accept in the present form